# Development of Free-Standing Titanium Dioxide Hollow Nanofibers Photocatalyst with Enhanced Recyclability

**DOI:** 10.3390/membranes12030342

**Published:** 2022-03-18

**Authors:** Nurul Natasha Mohammad Jafri, Juhana Jaafar, Farhana Aziz, Wan Norharyati Wan Salleh, Norhaniza Yusof, Mohd Hafiz Dzarfan Othman, Mukhlis A. Rahman, Ahmad Fauzi Ismail, Roshanida A. Rahman, Watsa Khongnakorn

**Affiliations:** 1Advanced Membrane Technology Research Centre (AMTEC), School of Chemical and Energy Engineering, Faculty of Engineering, Universiti Teknologi Malaysia, UTM Johor Bahru, Skudai 81310, Johor, Malaysia; nnatasha6@live.utm.my (N.N.M.J.); farhana@petroleum.utm.my (F.A.); hayati@petroleum.utm.my (W.N.W.S.); norhaniza@petroleum.utm.my (N.Y.); hafiz@petroleum.utm.my (M.H.D.O.); mukhlis@petroleum.utm.my (M.A.R.); afauzi@utm.my (A.F.I.); 2School of Chemical and Energy Engineering, Faculty of Engineering, Universiti Teknologi Malaysia, Johor Bahru 81310, Johor, Malaysia; r-anida@utm.my; 3Center of Excellence in Membrane Science and Technology, Department of Civil and Environmental Engineering, Prince of Songkla University, Songkhla 90110, Thailand; watsa.k@psu.ac.th

**Keywords:** photocatalysis, Bisphenol A degradation, vacuum filtration, titanium dioxide, hollow nanofibers, electrospinning

## Abstract

Titanium dioxide hollow nanofibers (THN) are excellent photocatalysts for the photodegradation of Bisphenol A (BPA) due to their extensive surface area and good optical properties. A template synthesis technique is typically employed to produce titanium dioxide hollow nanofibers. This process, however, involves a calcination procedure at high temperatures that yields powder-form photocatalysts that require post-recovery treatment before recycling. Meanwhile, the immobilization of photocatalysts on/into a membrane has been reported to reduce the active surface area. Novel free-standing TiO_2_ hollow nanofibers were developed to overcome those shortcomings. The free-standing photocatalyst containing 0.75 g of THN (FS-THN-75) exhibited good adherence and connectivity between the nanofibers. The recyclability of FS-THN-75 outperformed the THN calcined at 600 °C (THN-600), which retained 80% of its original weight while maintaining excellent degradation performance. This study recommends the potential application of free-standing TiO_2_ hollow nanofibers as high potential novel photocatalysts for the treatment of BPA in wastewater.

## 1. Introduction

Due to the health impacts that endocrine-disrupting compounds induce, their presence in wastewater has been a subject of concern. One of the identified compounds is Bisphenol A (BPA), which is a common monomer in a variety of industries, including the manufacturing of food containers, plastic bottles, and electronic equipment [1]. Previous research suggested that the intake of BPA into the human body might lead to reproductive issues, cardiovascular illness, and cancerous tumors [2].

Associated with the high resistance of organic pollutants, the existing conventional water-cleaning technologies such as membrane adsorption and coagulation are not capable of completely destroying these contaminants, resulting in high remaining concentrations released in treated effluents [3,4]. Furthermore, these technologies do not eliminate the pollutants but instead generate suspended sludge that requires after-treatment and disposal. Nowadays, photocatalytic degradation is gaining attention owing to its ability to oxidize a wide spectrum of organic compounds employing semiconductor photocatalysts such as titanium dioxide (TiO_2_).

Since photo-oxidation is a surface-dependent reaction, hollow-nanostructures are desired as a photocatalytic material [5,6,7,8]. TiO_2_ hollow nanofibers (THN) with high surface area, low bandgap energy, and superior photocatalytic performance to commercial TiO_2_ (Degussa P25) can be synthesized via template synthesis. However, at the end of the synthesis process, the photocatalyst needs to undergo a calcination procedure at a high temperature, typically above 500 °C, to remove the template and for amorphous-to-crystalline phase transition. Subsequently, this process often produces photocatalysts in powder form, which is employed in a suspended photocatalytic reaction system. This system suffers from several drawbacks: (1) low catalyst recyclability; (2) catalysts nanotoxicity. The small-sized photocatalysts tend to be trapped together in the effluent, which eventually reduces the number of remaining catalysts in the system. Catalysts reclamation procedures such as centrifugation [9], gravitational sedimentation [10], and filtration [11] are also required in order to collect the lost catalysts. In long-term application, this might have an impact on the operational cost and impede the overall performance as the recyclability of the photocatalysts are reduced. Other than that, the possible toxicity of TiO_2_ in the water effluent is also becoming a concern. Studies suggested that the continuous intake of TiO_2_ nanoparticles into the body system might cause oxidative stress and impact the intestinal mucosa, brain, heart, and other internal organs, increasing the risk of developing various illnesses, tumors, or the progression of existing cancer processes [12].

Although the immobilization of photocatalysts offers good attachment of the catalysts in/on a membrane/substrate, this would eventually inhibit the active surface area for light irradiation [13,14,15,16]. To the best of our knowledge, the assembly of TiO_2_ hollow nanofibers by vacuum filtration has not been reported elsewhere, although this approach is widely used as a facile and quick way to prepare free-standing carbon-based films [17,18,19]. Therefore, in order to improve the sustainability of photocatalytic oxidation technology and enhance its recyclability, in this work, we produced free-standing TiO_2_ hollow nanofibers (FS-THN) for the photodegradation of Bisphenol A. The THN was initially obtained in powder form as a result of the employed template synthesis approach. Then, the resulting THN was assembled into a free-standing THN sheet via facile vacuum filtration. The FS-THN exhibited good attachment and excellent recyclability, which are favorable for repeated cycles of BPA photodegradation.

## 2. Materials and Methods

### 2.1. Materials

Polyacrylonitrile, PAN at MW = 150,000, and titanium (IV) isopropoxide, TTIP, (97% in solution) were bought from Sigma Aldrich. N,N-dimethylformamide (DMF) was bought from RCI Labscan (Bangkok, Thailand). Meanwhile, acetic acid (glacial 100% purity) and nitric acid (65% purity) were bought from Merck (Darmstadt, Germany). All materials were used directly without any purification.

### 2.2. Synthesis of Titanium Dioxide Hollow Nanofibers

Titanium dioxide hollow nanofibers were obtained through template synthesis, as shown in Figure 1. Firstly, PAN nanofibers were prepared using 8 wt% of polyacrylonitrile, PAN MW = 150,000, Sigma Aldrich (Saint Louis, MO, USA) dissolved in dimethylformamide, DMF by RCI Labscan (Bangkok, Thailand). The dope solution was subjected to an electrospinning process conducted at the flow rate of 1 mL/h, 15 cm distance from needle tip to collector, and 12 kV voltage. The as-spun PAN nanofibers template was dried prior to the coating process. The PAN nanofibers were dip-coated in TiO_2_ sol-gel consisting of 10 mL of TTIP, 10 mL of acetic acid, 100 mL of distilled water, 1 mL of nitric acid, and HNO_3_ with the immersion and withdrawal speed of 5 mm/s. The PAN/TiO_2_ composite was dried in the oven at 90 °C for 2 h. Then, the nanofibers were calcined in air 400, 500, and 600 °C for 4 h. The samples were denoted as THN-400, THN-500, and THN-600, accordingly. 

### 2.3. Characterisation

The morphology of the nanofibers was observed using a field emission scanning electron microscope (FESEM, ZEISS Crossbeam 340, Jena, Germany). Phase identification of the crystalline material was made using an X-ray diffractometer (XRD, Rigaku D/Max 2200 PC, Tokyo, Japan) with CuKα radiation (λ = 1.540 Å, 40 kV and 30 mA). Nitrogen adsorption/desorption measurement was performed to examine the surface area of the nanofiber samples with an automatic gas adsorption instrument (BEL, Belsorp-max, Osaka, Japan). A UV-Vis-NIR Spectrophotometer (UV-3101PC Shimadzu, Kyoto, Japan) was used in this study to measure the optical absorption behaviors of the photocatalyst. 

### 2.4. Photodegradation Experiment

The photocatalytic ability of THN was evaluated by studying the photodegradation of bisphenol A (BPA) under UV light in a suspended photocatalytic reactor. The photocatalytic degradation of BPA was conducted under the illumination of a 3.0 mW/cm^2^ UV lamp (Vilber Lourmat, λ = 312 nm, 30 W) manufactured by Wuhan Co-shine Technology Co., Ltd., China, installed in a custom-made stainless steel photoreactor (60 × 35 × 42) cm^3^. The photoreactor consisted of a 1 L glass beaker, a magnetic stirrer, and a UV lamp positioned 15 cm above the beaker. The BPA concentration was quantified by using high-performance liquid chromatography (HPLC, Agilent Technology 1200 Series, Santa Clara, CA, USA) coupled with a programmable UV detector.

The photocatalysts were suspended in 10 mg/L of model wastewater BPA solution. The solution and photocatalyst were first dispersed using an ultrasonicator and then oxygenated in the dark for 1 h to reach an adsorption–desorption equilibrium. At every 15 min interval, 10 mL of sample aliquot was collected to analyze the adsorption percentage. After 1 h, the UV lamp was turned on while keeping the solution magnetically stirred. The sample aliquot was also collected at a regular time interval. The BPA concentration was measured using high-performance liquid chromatography (HPLC, Agilent Technology1200 Series). The degradation percentage was determined by Equation (1):(1)$$\mathrm{BPA}\;\mathrm{degradation}\;(\%)=\frac{C_{0}-C_{t}}{C_{0}}\times 100$$
where *C*_0_ and *C**_t_* are the initial BPA concentration and BPA concentration at time = t, respectively. 

### 2.5. Preparation and Performance of Free-Standing Photocatalysts

Free-standing THN were prepared by a simple vacuum filtration technique. A total of 0.50 g and 0.75 g of the as-prepared THN were dispersed in absolute ethanol (Merck). The solution was then poured into the Buchner funnel (diameter = 4 cm) and vacuum filtered for 10 min. The obtained photocatalyst was dried at room temperature for 24 h. Each sample was denoted as FS-THN-50 and FS-THN-75. 

In order to evaluate the performance of the synthesized FS-THN, the photocatalyst was observed for any cracks or breakage, and the one with the sturdiest structure was used in the photodegradation of BPA. In comparison, THN-600 was also used to evaluate the recyclability of both photocatalysts. The degradation efficiency was calculated by applying Equation (1). THN-600 that was in powder form was separated from BPA solution using centrifugation for 30 min. The supernatant was removed, and the remaining catalysts were collected to be used for the successive cycles. Both catalysts were washed with distilled water and ethanol to eliminate the impurities such as the unreacted reactants or by-products and dried. The experiment was then continued with a fresh batch of BPA solution.

The experiment was repeated for five cycles to study the recyclability of the photocatalysts. The performance of both catalysts was compared by evaluating the degradation efficiency and weighing the catalysts after each consecutive cycle to monitor the catalyst loss.

## 3. Results

### 3.1. Characterisation

The elemental composition of the nanofibers was studied through energy dispersive X-ray (EDX) result, as portrayed in Figure 2. Figure 2a,b shows the mapping and elemental composition of PAN/TiO_2_ nanofibers. The weight percentage of Ti, O, and C in PAN/TiO_2_ nanofibers were 35%, 55.1%, and 9.9%, respectively. After calcination, the composition of Ti, O, and C had changed to 52.1%, 44.9%, and 2.9%, as shown in Figure 2c,d. The decrease in the percentage of carbon indicates the decomposition of organics during the calcination process, thus resulting in hollow structured nanofibers.

The morphology of titanium dioxide hollow nanofibers can be observed from the FESEM images in Figure 3. Meanwhile, the crystallinity of the nanofibers was analyzed based on the XRD pattern in Figure 4. The change in calcination temperature has significantly influenced the morphology and crystallinity of the synthesized photocatalysts. For THN-400, THN-500, and THN-600, all samples exhibited fibrous morphology with decreasing diameters as the calcination temperature was increased. Apart from that, we can also observe the evolution of hollow structure formation with the rise in temperature, where the hollow structure is the most apparent in THN-600, and none was observed in THN-400.

The FESEM image of THN-400 showed the emergence of nanofibers lumen in the cross-section (Figure 3). However, due to the relatively lower temperature, the hollow nanofibers were not successfully formed due to the incomplete removal of the PAN template, and instead, a boundary between the lumen and outer shell was formed. At 400 °C, the nanofibers were crystallized to pure anatase phase. All of the XRD diffraction peaks of THN-400 are well-defined and agree well with anatase TiO_2_ (JCPDS card no. 21-1272). The appearing peaks at the positions of 2θ = 25.3°, 37.7°, 41.2°, 48.0°, 55.0°, 62.7°, and 75.0° correspond to crystal planes (101), (004), (112), (200), (211), (204), and (215).

The rise in the calcination temperature had induced the evolution of hollow structure formation and the anatase-to-rutile transition in the nanofibers. For THN-500, the hollow nanofibers structure was more prominent, and the rutile percentage in the nanofibers had increased to 17.9%. The appearing peaks at 2θ = 27.4°, 36.0°, and 68.7° correspond to crystal planes (110), (101), and (301) planes of the rutile TiO_2_ (JCPDS card no. 21-1276). The rather weak peaks imply a minute quantity of the rutile phase. This phenomenon was similar to the result that reported that the transformation of hollow NiO nanostructures in terms of morphology and crystallinity was highly influenced by the change in calcination temperature [20]. 

Meanwhile, the FESEM image of THN-600 revealed a hollow nanofibers structure with a rougher surface. This condition was in accordance with the more prominent rutile peaks, which implies higher crystallinity in the XRD spectrum of THN-600 with a composition of 75.8% rutile and 24.2% anatase [21]. Additional rutile peaks appear at 2θ = 41.2°, 56.6°, 56.8°, 62.8°, and 69.7°, corresponding to the planes (111), (211), (220), (002), and (112). The sharpness and intensity of the existing rutile peaks in the (110), (101), and (301) planes had increased. Successful formation of TiO_2_ hollow nanofibers depends on: (1) the complete decomposition of PAN template and (2) the sol-gel crystallization of TiO_2_. Based on the results obtained, 600 °C was optimum for simultaneous crystalline TiO_2_ formation and the complete removal of the PAN component, resulting in the formation of TiO_2_ hollow nanofibers. Based on the diameter distribution, the reported diameter of THN-400, THN-500, and THN-600 are 240 ± 39 nm, 218 ± 33 nm, and 177 ± 32 nm, accordingly. The decreasing trend of the nanofiber diameter is mainly ascribed to the shrinkage during the heating process.

Figure 5 illustrates the nitrogen adsorption–desorption isotherm for the THN calcined at various temperatures. All of the isotherm plots obtained exhibit Type II isotherm. The quantity of adsorbed nitrogen increases for the sample calcined at a higher temperature. The BET surface area of THN-400 was found to be 13.322 × 10^3^ m^2^/kg (Table 1). After increasing the calcination temperature to 500 °C, the surface area increased to 43.499 × 10^3^ m^2^/kg, which closely agreed to the value reported for commercial TiO_2_, Degussa P25 [22]. The surface area increased even further when the calcination temperature was set at 600 °C. The pore volume also exhibited the same trend as the specific surface area. Hence, referring to the FESEM result (Figure 3), the presence of hollow cavities and rougher surfaces in the THN were considered as the reason for the improvement of the photocatalyst surface area [23,24,25,26]. In comparison to the other adsorption–desorption isotherms, BET isotherm includes multilayer adsorption and continues to be the most widely used method to analyze the physical adsorption of gas molecules on a solid surface [27].

Figure 6 shows the UV-Vis absorption spectra of the THN. It can be seen that the maximum absorbance is observed for THN-600. The bandgap energy of the photocatalysts was estimated using the Kubelka–Munk function. A graph of (ahv)^1/2^ against the energy of absorbed light (eV) was plotted, as shown in Figure 6b. The bandgap energy was found to decrease from 3.50, 3.24, and 3.00 eV as the calcination temperature increased. These results indicate that THN-600 has better optical properties than THN-400 and THN-500. The absorption wavelength for THN-400, THN-500, and THN-600 was 381, 403, and 422 nm, respectively. Based on the Tauc plot, the values of the band gap energy obtained for THN-400, THN-500, and THN-600 are 3.50, 3.24, and 3.00 eV, respectively. It can be deduced that the band gap value is influenced by the morphology as well as the crystallinity of the THN. The band gap energy decreases with an increase in the rutile percentage, as the reported band gap for rutile is intrinsically lower than that of anatase [28].

### 3.2. Photocatalytic Degradation Performance

A preliminary degradation study was first conducted using all the prepared catalysts (THN-400, THN-500, and THN-600) to evaluate their photocatalytic performance. Then, the experiment was proceeded using the photocatalyst with the highest degradation to determine the optimum photocatalyst dosage. 

Figure 7 shows the BPA degradation percentage over degradation duration for all photocatalysts. Despite the absence of light source in the first hour of reaction, THN-400, THN-500, and THN-600 were able to reduce 7.8%, 12.4%, and 13% of BPA, respectively. The low degradation value was attributed to the adsorption of BPA molecules onto the surface of THN instead of photodegradation of the pollutant. This is in agreement with the BET result, where THN-600 possessed the largest active surface area adsorbed the highest percentage of BPA. Upon the irradiation of UV light, a similar trend was shown by the nanofiber photocatalysts. After 4 h, 26.5%, 42%, and 71.5% of BPA were successfully degraded by THN-400, THN-500, and THN-600, respectively. There are several factors that influence the degradation percentage of BPA. Firstly, the larger surface area of THN-600 provides more active sites for BPA adsorption and degradation. Then, the degradation performance was also influenced by the optimal anatase-rutile ratio creates heterojunction between the two phases, which lowers the band gap energy and facilitates the charge carrier generation efficiency.

Therefore, to determine the optimum dosage of photocatalysts, the experiment was continued using THN-600, and the amount was varied from 0.25, 0.50, 0.75 to 1.0 g/L in 10 mg/L of BPA solution. The determination of the optimum catalyst amount is essential in any photocatalysis process. For the catalyst recycling process, the number of catalysts used in the next consecutive cycle should be close to the range of the pre-determined optimum dosage to maintain the excellent photocatalytic performance. Figure 8 demonstrates the BPA degradation using a varied amount of THN-600 for 5 h. In the dark condition, the concentration of the reduced BPA was non-significant. Hence, the reduction was attributed to the surface adsorption of BPA molecules. In the presence of UV light, the degradation percentage for the different catalyst dosage increased in this order: 0.25 g/L (51.8%) < 0.50 g/L (71.5%) < 1.0 g/L (77.1%) < 0.75 g/L (92.9%). With a smaller dosage, the degradation percentage increased as a consequence of the increasing number of active sites, which motivates the generation of hydroxyl radicals (•OH) that attacks the BPA molecules. However, beyond 0.75 g/L, a significant drop in the degradation percentage was observed. This can be explained by the light scattering effect that occurs when the THN-600 is dispersed in the BPA solution. This is similar to the reported decline in the degradation rate of malachite green using TiO_2_ nanoparticles that were assumed to be due to the light scattering and reduced light penetration with higher catalyst loading [29].

### 3.3. Assembly and Recyclability of Photocatalysts

For a heterogeneous photocatalytic system, in which the photocatalysts and the reactants exist in different phases, the recyclability of the photocatalysts is a critical aspect. Ideally, after the reaction, the catalysts return to their initial state without any deterioration. In order to regard that a catalyst has excellent recyclability, the catalysts need to be easily recovered and have stable efficiency throughout prolonged or repeated cycles of reaction. Photocatalysts in powder form have upper hands in terms of the surface area. However, the small-sized catalysts have the potential to be trapped in the effluent, causing marginal catalysts loss. On the other hand, photocatalysts films are more favored because they can be easily separated from the solution.

The optimum catalysts dosage of THN-600 has been determined to be at 0.75 g/L, which obtained a photodegradation efficiency of 92.9%. Firstly, 0.75 g of THN-600 was utilized in the synthesis of free-standing TiO_2_ hollow nanofibers. This sample was denoted as FS-THN-75. Figure 9 displays the assembled photocatalysts. After being peeled off, the THN-600 on the FS-THN-75 sheet firmly adheres, and no wrinkles or cracking and chipping can be observed on its surface. Attributed to the adequate amount of THN-600 used, as well as the difference in pressure between the inside and outside of the vacuum flask, the FS-THN-75 consisted of tightly stacked and firmly adhered THN-600. Then, the amount of THN-600 was reduced to 0.50 g to form a free-standing TiO_2_ hollow nanofibers sheet, which is denoted as FS-THN-50. The obtained FS-THN-50 is shown in Figure 9c. The image revealed that the surface of the photocatalysts tends to crack when peeled off from the filter paper. With less amount of THN-600, the nanofibers were loosely assembled and inadequately adhered, causing the FS-THN-50 to chip and break into smaller pieces.

Hence, we proceed with the recyclability experiment using FS-THN-75 as the photocatalyst. The recyclability of THN-600 and FS-THN-75 was evaluated by means of the final BPA removal percentage (after 1 h of absorption-desorption and 4 h of UV irradiation) under repeated cycles, and the results are presented in Figure 10. The weight of catalysts used was also measured prior to every cycle. During the first cycle, the degradation efficiency of THN-600 was 97.3%, which is 0.5% higher than that of FS-THN-75, despite having the same weight of catalysts. This slight difference is caused by the stacking of photocatalysts in FS-THN-75, which may hinder some parts of the active surface areas. As a result, fewer BPA molecules were adsorbed and mineralized on the surface of FS-THN-75. However, there is no drastic decrease in the photocatalyst efficiency, suggesting that the photocatalytic properties of THN-600 remain even after the modification into a free-standing form.

As shown in Figure 10, a significant decline in the weight and degradation efficiency of THN-600 can be observed in the following cycles. In the second, third, fourth, and fifth cycles, the degradation percentage of BPA was 85.6%, 73.95%, 62.7%, and 54.5%, respectively. This is in strong agreement with the weight loss of catalysts. The amount of remaining catalyst reduced from 0.75 g to 0.66 g, 0.57 g, 0.48 g, and 0.41 g from the first to fifth cycle. Although catalyst recovery has been made through centrifugation, the THN-600 still has low catalyst recovery due to its very minute size. The THN-600 catalyst might be collected with the sample aliquot during the experiment. This issue of considerable escape of photocatalysts with the effluents has been addressed by some previous [30,31]. On the other hand, the catalyst recyclability of FS-THN-75 was significantly improved. In the second, third, fourth, and fifth cycles, the weight of the remaining FS-THN-75 was 0.72 g, 0.69 g, 0.66 g, and 0.60 g, respectively. After five repeated cycles, FS-THN-75 is able to retain 80% of its original weight without requiring additional catalyst reclamation procedures. The small amount of catalyst loss might be caused by the leaching of some catalysts from the sheet. It should be noted that future work needs to be performed to further reduce the catalyst leaching, for example, through the study on the effect of the vacuum filtration duration or the drying temperature.

As discussed above, this implies that the modification of THN-600 into FS-THN-75 not only retains higher efficiency of BPA degradation compared to suspended THN-600 but also improves the recyclability. Taking the operation of an industrial wastewater plant, the simplicity of FS-THN-75 separation from the reaction medium, and the potential for long-term use can significantly reduce the cost and time of operation.

The photocatalytic degradation of BPA was also investigated by HPLC/MS to determine the intermediate products formed in this reaction. The HPLC retention time (T_R_) for each of the standard compounds is listed in Table 2. 

Apart from the BPA peak, three other compounds peaks were observed at shorter retention periods than the BPA peak. The BPA peak emerges at 16.4 min, while the peaks of the other components appear with retention times of 13.5, 12.8, and 5.2 min. The present analytical results of BPA intermediate products are generally in line with those previously published. In the degradation of BPA, aromatic compounds such as 4-isopropanolphenol, p-hydrobenzoic acid, 4-isopropenylphenol, and hydroquinone have been identified [32,33,34,35]. As a result of further degradation, short-chain aliphatic compounds such as formic acid, acetic acid, and benzaldehyde are also generated [36,37]. Therefore, a possible degradation mechanism of BPA was proposed, as presented in Figure 1. In general, the photocatalytic oxidation of BPA is initiated via attacks by •OH at the electron-rich C3 in the phenyl group of the BPA. Then, this is followed by the cleavage of the phenyl groups, forming 4-isopropanolphenol and phenol. Hydroquinone is produced from the oxidation of 4-isopropanolphenol. These compounds are then further degraded into short-chain aliphatic compounds through ring-opening reactions by the •OH radicals. Eventually, these compounds are mineralized into carbon dioxide and water.

## 4. Conclusions

The assembled THN (FS-THN) showed good attachment and interconnectivity. For the first cycle, the FS-THN-75 showed slightly lower BPA degradation than THN-600, most probably due to the stacking of photocatalysts that hindered some of its surface areas. However, the FS-THN-75 portrayed superior properties to the THN-600 in terms of recyclability. The FS-THN-75 is able to retain 80% of the catalyst weight while showing no significant decrease in the degradation efficiency. The ease with which FS-THN-75 may be separated from the reaction media, as well as the potential of long-term usage, can significantly decrease operating costs and time. In future works, it is suggested that the effect of vacuum filtration duration or the drying temperature is further studied to minimize the catalyst loss. The developed FS-THN-75 resolved the technical limitations of the synthesized TiO_2_ hollow nanofibers.

## Data Availability

The data presented in this study are available in https://doi.org/10.3390/membranes11080581 (accessed on 11 February 2022).

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
