# Peer review of "Development of Free-Standing Titanium Dioxide Hollow Nanofibers Photocatalyst with Enhanced Recyclability"

_membranes, 2022, doi:10.3390/membranes12030342_

Round 1

Reviewer 1 Report

The main objective of this paper is to assemble the synthesized TiO2 hollow nanofibers into a free-standing form applying chemical treatment and vacuum filtration techniques. In my opinion, this manuscript is interesting and the presented results are clear and sound. To enrich the manuscript, some comments are presented as follows. I recommend acceptance of manuscript after major revision:
1) What is the novelty of this manuscript compared to other published articles? please highlight them in the abstract and last paragraph of introduction
2) What are the advantage of BET isotherm compared to other adsorption-desorption isotherm?
3) Table 1: the units and quantities must be presented in SI.
4) If possible, the authors must bring the diameter, porosity and other related parameters of hollow nanofibers.
5) English of the manuscript needs revision
6) in the keywords section, the first letter must be written capital

Author Response

1) What is the novelty of this manuscript compared to other published articles? Please
highlight them in the abstract and last paragraph of introduction.
Response: The novelty of this study relies on the approach of assembling the hollow TiO2
hollow nanofibers powder into a free-standing form. Vacuum filtration technique has been
widely used to form carbon-based film, but the application of this technique for TiO2 hollow
nanofibers has not been done elsewhere. We have emphasized the novelty of this study in the
abstract (Line 22-23) and last paragraph of introduction (Line 67-69).
2) What are the advantage of BET isotherm compared to other adsorption-desorption
isotherm?
Response: We have addressed the advantage of BET isotherm in Line 220-222.
“In comparison to the other adsorption-desorption isotherms, BET isotherm includes multilayer
adsorption and continues to be the most widely used method to analyse the physical adsorption
of gas molecules on a solid surface [27].”
3) Table 1: the units and quantities must be presented in SI.
Response: The units and quantities have been changed to SI standard.
4) If possible, the authors must bring the diameter, porosity and other related
parameters of hollow nanofibers.
Response: The discussion on diameter of the nanofiber has been included (Line 199-202)
5) English of the manuscript needs revision
Response: We have corrected the grammatical and sentence structure mistakes in the
manuscript
6) In the keywords section, the first letter must be written capital
Response: The letters have been changed to capital letter

Reviewer 2 Report

The paper with title“development of freestanding titanium dioxide hollow nanofibers photocatalyst with enhanced recyclability” was impressive to me, it could be a new way to solve the issues on nanoparticle agglomeration resulting lower efficiency of photocatalysts. However, there are still questions and errors to be addressed.

  1. The synthesis of THN should be more clear.
  2. The lines color in Fig 4 and Fig 5 should be uniformed accordingly.
  3. Elemental analysis should be carried out by EDAX.
  4. Comparison between your work and others should be given.
  5. Should deeply investigate the mechanism of degradation of THN film.

Author Response

1) The synthesis of THN should be more clear.
Response: We have included a diagram for clearer explanation on the synthesis of THN (Figure
1)
2) The lines color in Fig 4 and Fig 5 should be uniformed accordingly.
Response: The color for data in Figure 4 and 5 (Figure 6 and 7 in the new version of the
manuscript) has been changed to similar color
3) Elemental analysis should be carried out by EDAX.
Response: We have provided elemental analysis discussion from EDX result (Line 153-159)
4) Comparison between your work and others should be given.
Response: We have included some comparison with previous literature to support our result.
For example: reference 31-35.
5) Should deeply investigate the mechanism of degradation of THN film.
Response: The mechanism of BPA degradation using FS-THN film has been discussed (Line
349-364) and Scheme 1

Reviewer 3 Report

The paper entitled “Development of Free-standing Titanium Dioxide Hollow Nanofibers Photocatalyst with Enhanced Recyclability” is an interesting work on the development and application of TiO2 hollow nanofibers for the photodegradation of organic compounds. The work is interesting and of great value. English could be improved. Other specific comments are reported below:
ABSTRACT
1)    There are some English errors, not only in the abstract but throughout the manuscript. It must be corrected. Example: Line 22: “Comparatively, he recyclability of FS75-THNFs …”, - is it “he” or “the”?
KEYWORDS
2)    I suggest changing it, as most of the keywords selected are already on the Title.
INTRODUCTION
3)     Line 45: “superior photocatalytic performance…” – Superior than what?
4)     Lines 62-64: Please add some study about this comparison (i.e.: 10.1016/j.cej.2019.122114)
RESULTS AND DISCUSSION
5)    Lines 234-236: This explanation is a bit vague. Please, explain more about how the larger surface area and crystallinity promoted higher BPA degradation. There are many other factors that may affect, such as: porosity, electrostatics, …
6)    Line 254-256: The effect is the same, however, is a different photocatalyst and a different organic compound. There are many references in the literature explaining the light scattering effect when using TiO2 and endocrine-disrupting compounds. I suggest changing the reference.
7)    Lines 318-322: Authors need to improve this part. The catalyst leaching is really significant (20% in 5 cycles). This aspect is a huge drawback for the technique used. There are many studies about the immobilization of TiO2 on supports where it is observed lower or no catalyst leaching within the cycles. So, authors must explain the advantage of the presented technique; suggestions on how to improve the procedure in order to minimize the leaching; comparison with other studies; etc..
CONCLUSIONS
8)    The latter comment is also valid for the conclusions.
9)    Line 328: “only a 5% reduction in the catalyst weight after five cycles…” This is incorrect. Please re-calculate this value and re-formulate your conclusion.

Author Response

Dear reviewer,
We appreciate your precious time in reviewing our paper and providing
valuable comments. The authors have carefully considered the comments and
tried our best to address every one of them. We hope the manuscript after careful revisions meet your high standards. The point-by-point responses are provided in the attechment.

Thank you.

Round 2

Reviewer 1 Report

the authors have revised the manuscript based on my comments. I recommend the acceptance of this manuscript

Reviewer 3 Report

The authors improved the manuscript accordingly.